# tagE: Enabling an Embodied Agent to Understand Human Instructions

**Chayan Sarkar** and **Avik Mitra** and **Pradip Pramanick** and **Tapas Nayak**
TCS Research, India
{sarkar.chayan, mitra.avik1, pradip.pramanick, nayak.tapas}@tcs.com

## Abstract

Natural language serves as the primary mode of communication when an intelligent agent with a physical presence engages with human beings. While a plethora of research focuses on natural language understanding (NLU), encompassing endeavors such as sentiment analysis, intent prediction, question answering, and summarization, the scope of NLU directed at situations necessitating tangible actions by an embodied agent remains limited. The inherent ambiguity and incompleteness inherent in natural language present challenges for intelligent agents striving to decipher human intention. To tackle this predicament head-on, we introduce a novel system known as **t**ask and **a**rgument **g**rounding for **E**mbodied agents (tagE). At its core, our system employs an inventive neural network model designed to extract a series of tasks from complex task instructions expressed in natural language. Our proposed model adopts an encoder-decoder framework enriched with nested decoding to effectively extract tasks and their corresponding arguments from these intricate instructions. These extracted tasks are then mapped (or grounded) to the robot's established collection of skills, while the arguments find grounding in objects present within the environment. To facilitate the training and evaluation of our system, we have curated a dataset featuring complex instructions. The results of our experiments underscore the prowess of our approach, as it outperforms robust baseline models.

## 1 Introduction

Robots in our daily surroundings are often engage with human beings for various purposes. As natural language interaction capability increases the acceptability and usability of a robot, many studies have focused on natural language interaction with a robot (Williams et al., 2015). This can be particularly useful if we can provide task instruction in natural language (Pramanick et al.,

2020). However, large vocabulary and variation of word/phrases/sentences of any natural language (e.g., English) makes it very difficult for a robot to understand human intention and perform the task (Pramanick et al., 2019b).

Recent advances in natural language processing (NLP), in particular, the rise of large-scale neural language models have simplified NLP tasks with high accuracy (Devlin et al., 2019; Brown et al., 2020). However, in order for a robot to perform a task, the task intention has to be mapped to a known set of skills of the robot (task grounding) so that some action can be taken in the physical world. Additionally, the objects associated with a task (task arguments) should be mapped to the objects within the environment (argument grounding). Most robots use an object detector, which uses a fixed vocabulary. A human may not be aware of that vocabulary or may not remember it. As a result, a different word/phrase can be used to refer the same object. Therefore, argument grounding becomes equally important in order to perform physical action in the environment.

Existing works on intent prediction map the intended task to the robot's capability (Brawer et al., 2018). However, they can neither extract the arguments associated with a task nor handle complex instructions with multiple tasks. Works on relation extraction generally find the relevant triplets in a natural language text, where the triplets have the form of head, relation, and tail $< h, r, t >$ (Nayak and Ng, 2019). One can use such a method to find a task-argument triplet (Figure 1a). But, multiple arguments may be associated with the same tasks. Extracting such triplets where the head/tail is common in many such triplets is not straightforward (Figure 1b, 1c). Moreover, the existing methods neither ground the task nor ground the arguments. For example, in Figure 1b, the word 'keep' signifies the 'PLACING' task according to the robot's capability, and the word 'fridge' needs

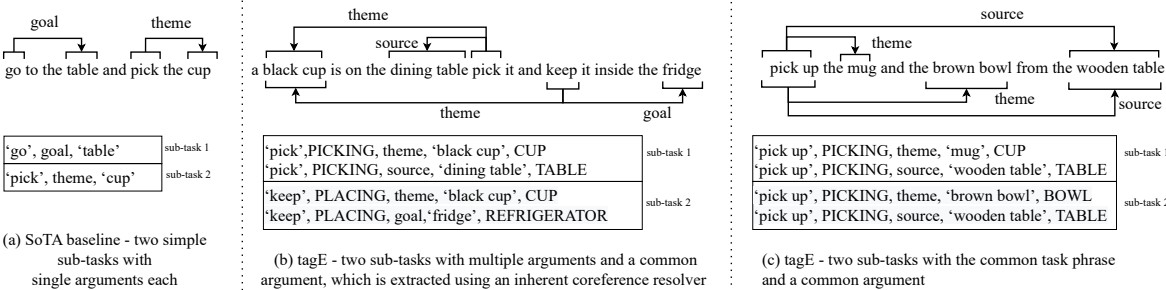

Figure 1: Some natural language task instruction – (a) example where existing triplet extractor can be employed for task and argument pair extraction, (b) example of why it is necessary to ground task and argument along with co-reference resolution (pentuple extraction), (c) example of why generative approach for task and argument extraction is required.

to be mapped to 'REFRIGERATOR' according to the object detector's vocabulary. Similarly, the same task phrase 'pick up' as well as the same argument phrase 'wooden table' are shared by two tasks as shown in Figure 1c. This can only be possible if an NLU system can generate as many triplets as possible from the given natural language instruction without any limitation.

We have developed a novel deep neural network architecture, called tagE (**t**ask and **a**rgument **g**rounding for **E**mbodied agents) that extract the set of tasks and their relevant arguments from a natural language complex instruction. The contributions of this article are following.

- We proposed a novel encoder-decoder architecture for natural language understanding that employs layered decoding. Unlike existing triplet extractors, tagE finds pentuple $< h, h_g, r, t, t_g >$, where $h_g$ and $t_g$ represent the grounded values of the task ($h$) and argument ($t$), respectively.

- tagE employs a shallow auto-regressive decoder, which enables it to extracts as many grounded task-argument pairs as possible, resolve co-reference, and handle the shared task and/or argument phrase.

- We have curated a new dataset that enables grounded task-argument extraction. The model can ground the argument based on object detector vocabulary. The task-argument relation data and argument grounding data are detached in a way that the object detector as well as the class of objects can be expanded/modified easily[1].

---

[1] Any resource related to this work will be made available at https://github.com/csarkar/tagE.

## 2 Related Work

There are three different areas that are relevant to this work – sequence-to-sequence learning, structured information extraction, and natural language understanding of instructions for robots.

**Sequence-to-sequence learning:** Encoder-decoder models are popular for the sequence-to-sequence (S2S) learning. There are different types of S2S tasks for which encoder-decoder architecture is used, e.g., neural machine translation (Sutskever et al., 2014; Bahdanau et al., 2015; Luong et al., 2015), joint entity and relation extraction (Zeng et al., 2018; Nayak and Ng, 2020), cross-lingual open information extraction (Zhang et al., 2017), joint extraction of aspect-opinion-sentiment triplets (Mukherjee et al., 2021), etc.

**Structured Information Extraction:** Structured information extraction from unstructured text is an important task in natural language processing. Entity-Relation extraction (Miwa and Bansal, 2016; Shen and Huang, 2016; Vashishth et al., 2018; Nayak and Ng, 2019), aspect-sentiment triplets extraction (Xu et al., 2020; Jian et al., 2021; Mukherjee et al., 2021), causality extraction (Li et al., 2021), event extraction (Liu et al., 2018; Sha et al., 2018), attribute-value extraction (Roy et al., 2021, 2022) are such important tasks. 'BIO' tag-based sequence labeling models are popular to extract structure information from text. In recent times, pointer networks are explored for such tasks (Vinyals et al., 2015). Seo et al. (2017); Kundu and Ng (2018) used pointer networks for the machine-reading comprehension task to identify the answer span in the passage. Nayak and Ng (2020) used a similar network to identify the entities in sentences for joint entity and relation extraction tasks. Similarly, Becquin (2020) used such networks for identifying

casual spans in the text. (Mukherjee et al., 2021) used pointer networks for joint extraction of aspect-opinion-sentiment triplets from online reviews.

**Natural Language Understanding for Robots:** Natural language understanding for robots mostly involves executing human-provided directives given in natural language. Significant progress has been made by - i) restricting the action space of the robot to navigation, i.e., posing it as a Vision and Language Navigation (VLN) problem, and ii) providing detailed step-by-step instructions that either reduce or remove the burden of planning from high-level task goals (Anderson et al., 2018; Blukis et al., 2019; Shah et al., 2022). A few works have attempted to include manipulation in VLN, but still allowing step-by-step instructions and limited to a single or a constrained set of manipulation actions (Misra et al., 2018; Kim et al., 2020; Pashevich et al., 2021). We focus on a more general problem that assumes the arbitrary action capabilities of a robot that includes both navigation and manipulation. Thus the problem can be defined as generating an action sequence (plan) for a high-level natural language instruction that contains one or more tasks. Several approaches have been proposed to solve this. Predominant methods exploit the embodied nature of a robotic agent to infer and refine the plan by primarily using multi-modal input that includes visual feedback and action priors (Paxton et al., 2019; Shridhar et al., 2020; Singh et al., 2021; Zhang and Chai, 2021; Ahn et al., 2022). Thus natural language understanding in these systems is simplified by obtaining a latent representation of the language input to bias the inference using attention modeling. Embodied agents that can answer natural language queries have also been developed by following a similar approach of planning by biasing the agent's exploration using latent linguistic representations (Das et al., 2018).

Several works proposed models that perform visual grounding of referring expressions in an embodied agent's ego-view, following a similar approach for encoding language input (Qi et al., 2020; Rufus et al., 2021; Roh et al., 2022; Pramanick et al., 2022). A major limitation of this approach of end-to-end learning with multi-modal input is that the models are often heavily biased towards a specific dataset, simulator, and agent with specific capabilities. Thus, they often exhibit poor generalization to unseen environments and fails to generate plan for unseen composition of tasks (Shridhar

et al., 2020; Min et al., 2021). Though these models are particularly effective for following detailed instructions with explicitly mentioned tasks in known environments, they often generate incorrect plan for high-level instructions with implicit sub-goals that require long-horizon task planning (Min et al., 2021). Corona et al. (2021) proposed segmenting the instruction into a sequence of task types and training separate seq2seq models for different task types. Subsequently, several works have proposed modular methods that decouple multi-modal seq2seq learning into separate language understanding, perception and controller components. For example, Nguyen et al. (2021) proposed a model with separate decoders for predicting task predicates of navigation and manipulation parts of the instruction. Similarly, other works (Jansen, 2020; Huang et al., 2022) explored plan generation using large pre-trained language models. Liu et al. (2022) trained two separate models for action and object prediction, and Ri et al. (2022) studied plan generation solely from language input by training a CTC-based model. However, all of these models directly predict low-level action sequence or grounded goal predicates; thus still learns agent and environment specific representations.

Existing approaches of structured prediction from instructions follow a sequence labeling approach which has two major limitations - i) it can't classify tokens multiple times and/or with separate class label, and ii) it can't handle co-reference in complex instructions. In contrast, we propose a novel generative model that has significant advantages over sequence labeling.

## 3 Proposed Framework

In this section, we formally define the task before describing the neural network architecture. Given a natural language instruction $X = \{x_1, x_2, ..., x_n\}$ with $n$ tokens, the goal is to extract a set of tasks, $T = [t_j | t_j = (t_j^s, t_j^e, t_j^l)]_{j=1}^{|T|}$, where $t_j$ is the $j$'th task, $|T|$ is the number of tasks, $t_j^s$ and $t_j^e$ represents the positions of the start and end tokens of the task description span, and $t_j^l$ represents the type of the task (grounded task). Additionally, we extract the set of arguments for each task. Specifically, for task $t_j$, we extract $A_j = [a_{jk} | a_{jk} = (a_{jk}^s, a_{jk}^e, a_{jk}^l)]_{k=1}^{|A_j|}$ where $a_{jk}$ is the $k$'th argument, $|A_j|$ is the number of arguments for the task, $a_{jk}^s$ and $a_{jk}^e$ represents the positions of the start and end tokens of the argument description span, and $a_{jk}^l$ represents the

| index | 0 | 1 | 2 | 3 | 4 | 5 | 6 | 7 | 8 | 9 | 10 | 11 | 12 | 13 | 14 | 15 |
|-------|---|---|---|---|---|---|---|---|---|---|----|----|----|----|----|----|
| token | the | black | cup | is | on | the | dining | table | pick | it | and | keep | it | inside | the | fridge |
| annotation | 3 3 Being_located, 1 2 Theme, 6 7 Source ;
8 8 Picking, 1 2 Theme, 6 7 Source ;
11 11 Placing, 1 2 Theme, 15 15 Goal, 15 15 Container object | | | | | | | | | | | | | | | |
| BIO tag | O | B_cup | I_cup | O | O | O | B_table | I_table | O | O | O | O | O | O | O | B_refrigerator |

Figure 2: Example of instruction annotation - the annotation for grounded task types and the corresponding argument types are shown in row 3, and the BIO tags for argument (object) grounding is shown in row 4.

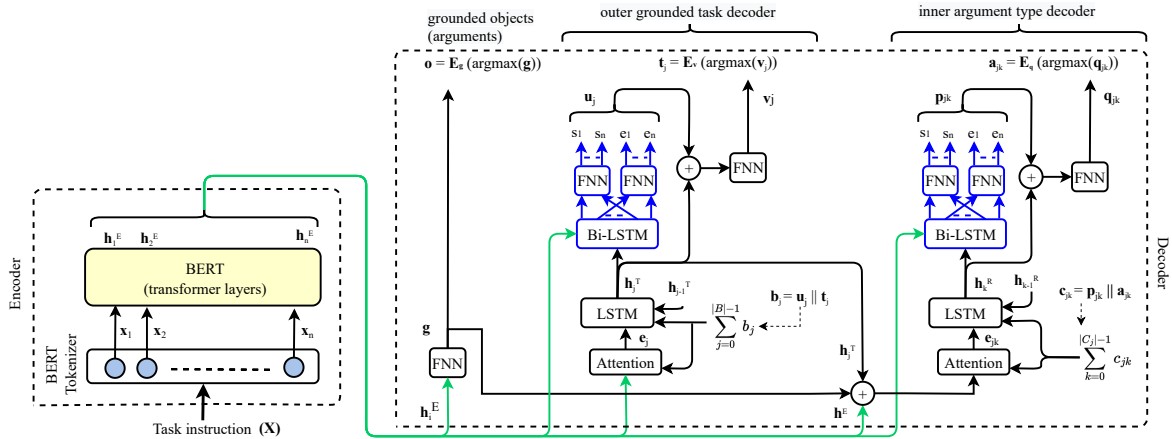

Figure 3: Encoder-decoder architecture of our tagE model.

type of the argument. The number of arguments for the tasks can be different. We added the example of such positional sequence in Figure 2. Additionally, we do the grounding of the arguments, i.e., map the span of an argument to an object if it is a physical object.

In Figure 3, we give an overview of our proposed model. We use an encoder-decoder model inspired from Nayak and Ng (2020); Mukherjee et al. (2021) for this task. The encoder is used to encode the given natural language instruction and decoder is used to generate the task and argument sequence. However, Nayak and Ng (2020) proposed a linear decoding process for joint entity and relation extraction task and Mukherjee et al. (2021) proposed an aspect-sentiment triplets extraction task. But such linear decoding scheme is not suitable for this task. We need to extract a list of tasks and for each task, we need to extract a list of arguments. Thus, we propose a novel nested decoding approach for task and argument extraction. The outer decoder is responsible to generate the task sequence and inner decoder is responsible to generate the argument sequence for each task. For argument grounding, we use a 'BIO' sequence labeling based approach. We jointly train our model for these three objectives of

task extraction, argument extraction, and argument grounding together.

### 3.1 Instruction Encoder

We use pre-trained BERT tokenizer and BERT model (Devlin et al., 2019) to obtain the contextual encoder representation of the input tokens, $\mathbf{h}_i^E \in \mathbb{R}^{d_h}$, where $\mathbb{R}^{d_h}$ is the vector of real numbers of dimension $d_h$.

### 3.2 Object Grounding

The arguments that refers to a physical object in the environment needs to be grounded, i.e., mapped to the object detector's vocabulary that the agent is using. We use 'BIO' sequence labeling approach for object grounding. The encoder vectors $\mathbf{h}_i^E$ are passed to a feed-forward layer with softmax activation for classifying a span to grounded objects. From this sequence labeling, we obtain the list of object spans in the instruction along with the grounded object class. Each argument span extracted by the inner argument decoder is assigned to the class type from this list. In our 'BIO' tagging scheme total number of tags, $K = 2 \times \#grounded\_objects + 1$. $\#grounded\_objects$ is the number of objects in the vocabulary of the object detector of the the

robot. For each token in the instruction, we get $K$ dimensional softmax output vector. We use this vectors in our argument extraction network to improve the extraction process. Please note that for object (argument) grounding, sequence labeling is sufficient as even if an argument is shared in multiple tasks, it is detected as the same object by the object detector.

### 3.3 Task Sequence Generation

We use an LSTM-based auto-regressive decoder for generating the task sequence. At every time step $j$ of the task decoding, decoder LSTM receives an overall representation of encoder context $\mathbf{e}_j \in \mathbb{R}^{d_h}$ and representation of the previously generated tasks ($\sum_{j=0}^{|B|-1} \mathbf{b_j}$) as input. The encoder context representation $\mathbf{e}_j$ is obtained using an attention mechanism on encoder hidden states $\mathbf{h}_i^E$ as used in Bahdanau et al. (2015) for neural machine translation. At the first decoding step, $\mathbf{b_0}$ is taken as a zero vector, and the subsequent task vectors $\mathbf{b_j}$'s are obtained after each decoding step (described in Section 3.5). Task decoder LSTM outputs the task hidden vector $\mathbf{h}_j^T \in \mathbb{R}^{d_h}$. This hidden representation is concatenated with the hidden representations $\mathbf{h}_i^E$ of the encoder and passed to a span detection module to mark the start and end token of the task description. This span detection module is described later in detail. After one task is extracted, the inner decoder (argument decoder) receives the task hidden vector $\mathbf{h}_j^T$ and starts generating the argument sequence for this task. Once all the arguments of this task are generated, this outer task decoder moves on to generate the next task.

### 3.4 Argument Sequence Generation

Like task decoding, we use a similar LSTM-based auto-regressive decoder for extracting the arguments. For the extracted task $\mathbf{t_j}$, at every time step $k$ of the argument decoding process, the argument decoder LSTM receives the task representation vector $\mathbf{h}_j^T$, encoder context $\mathbf{e}_{jk}$ vectors, and all the previously generated arguments for this task ($\sum_{k=0}^{|C_j|-1} \mathbf{c_{jk}}$).

### 3.5 Span Identification

We identify the task and argument description spans in the instruction using their start and end tokens. We use two different span detection modules (shown in blue color in Figure 3) for task and argument span identification. But the design of

---

**Algorithm 1** Proposed nested decoding algorithm

TD = init_task_decoder
AD = init_argument_decoder
**for** $j$ in $range(max\_task\_cnt)$ **do**
    TD decodes a task $t_j$
    **for** $k$ in $range(max\_arg\_cnt)$ **do**
        AD decodes an argument $a_{jk}$ for task $t_j$
    **end for**
**end for**

---

these two modules is similar in nature. They have a BiLSTM layer with two feed-forward layers with softmax activation. We concatenate the task hidden vector $\mathbf{h}_j^T$ or argument hidden vector $\mathbf{h}_k^A$ with the encoder vectors $\mathbf{h}_i^E$ and pass it to the BiLSTM layer. The output of the BiLSTM layer is passed to a feed-forward layer to convert each hidden representation to a scalar score. We get a scalar score corresponding to each token in the instruction. We apply softmax activation across these scalar scores to normalize them. The corresponding token with the highest normalized score is marked as the start token of a span. Similarly, another feed-forward layer with softmax activation is used to mark the end token of the span. We obtain the vector representations of these spans ($\mathbf{u_j}$ or $\mathbf{p_{jk}}$) using the normalized scores and BiLSTM outputs as done in Nayak and Ng (2020).

Next, $\mathbf{u_j}$ and $\mathbf{h}_j^T$ are concatenated and passed to a feed-forward layer with softmax to classify the task type ($\mathbf{v}_j$). We use $\mathbf{v}_j$ and a task type embedding ($\mathbf{E}_v$) layer to get the task type vector $\mathbf{t}_j$. Further, the vector representation of the task $\mathbf{b}_j$ is obtained by concatenating $\mathbf{u_j}$ and $\mathbf{t}_j$. Similarly, we classify the argument type ($\mathbf{a}_{jk}$) and obtain the vector representation of the argument $\mathbf{c}_{jk}$ as shown in Figure 3.

### 3.6 Training and Inference

We train our proposed model in mini-batches of size $B$ for multiple epochs and update the model parameters using the negative log-likelihood loss and gradient descent-based optimizer AdamW (Loshchilov and Hutter, 2019). Following is the loss function for the task extraction decoder.

$$\mathcal{L}_t = -\frac{1}{|B|} \sum_{j=1}^{|B|} [\ln(s_j) + \ln(e_j) + \ln(c_j)]$$

where $s$ and $e$ are the softmax outputs of the corresponding gold-label start and end positional index

of the task span and $c$ is softmax output of the gold-label task type.

Similarly, the following is the loss function for the argument extraction decoder.

$$\mathcal{L}_a = -\frac{1}{\sum_{j=1}^{|B|}|C_j|}\sum_{j=1}^{|B|}\sum_{k=1}^{|C_j|}[\ln(s_{jk}) + \ln(e_{jk}) + \ln(r_{jk})]$$

where $s$ and $e$ are the softmax outputs of the corresponding gold-label start and end positional index of the argument span and $r$ is softmax output of the gold-label argument type.

Following is the loss for the argument grounding.

$$\mathcal{L}_g = -\frac{1}{n}\sum_{i=1}^{n}\ln(g_i)$$

where $g$ is softmax output of the gold-label tag for the $i$-th token in the instruction. The final loss for a mini-batch of size $M$ of the tagE model is calculated as follows.

$$\mathcal{L} = \frac{1}{M}\sum_{m=1}^{M}[\mathcal{L}_t + \mathcal{L}_a + \mathcal{L}_g]$$

During training, we include the end of sequence 'EOS' task and 'EOS' argument at the end of their sequences and let the model learn to stop extraction of tasks/arguments. At the inference time, we run both decoders for a maximum number of steps but ignore any tasks/arguments extracted after the 'EOS'. During inference, we follow a greedy approach to select the start and end tokens of the task or argument spans. We select the start and end where the product of the two probabilities is maximum, and the end token does not appear before the start token. We include the parameter settings used to train our tagE model in Table 1.

| Batch size | 16 |
| Optimizer | AdamW |
| Learning rate | 0.0001 |
| #Epochs | 100 |
| Early stop count | 20 |

Table 1: Parameter settings used to train the tagE model.

## 4 Experiments

In this section, we describe the dataset, parameter settings, and evaluation metrics that are used to train and evaluate our model.

### 4.1 Dataset

We annotate a new dataset for our experiments. We build the dataset by extending the natural language instruction sentences from two robotic commands datasets (Shridhar et al., 2020; Vanzo et al., 2020). The current set of task types is included in Table 8. The task types are also adapted from robotic task datasets. The first two tasks are not real task types, but often appear in robotic task instruction and need to be processed for understanding the other tasks. Though our dataset is adapted from the existing ones, the annotation is significantly different to suit the purpose. This consists of the following features – (i) the token span of the set of sub-tasks in the instruction along with their grounded types, (ii) the set of arguments associated with each task along with the token span and the argument type, (iii) provision of sharing an argument by multiple tasks, (iv) provision of classifying a token multiple times, (v) resolution of co-reference of objects, and (vi) argument grounding considering the object classes of the underlying object detector.

Figure 2 depicts our annotation schema with an example. As any task type or argument type can be represented by multiple tokens in the input, we annotate each type with a start and end token. For example, the 'Source' of the cup is denoted by two tokens 'dining table', which is marked by the start and end token index 6 and 7, respectively. There are three sub-tasks in the example – 'Being_locatede', 'Picking', and placing. Each sub-task and its corresponding arguments form a substructure, which is separated by a vertical line as shown in the example. Within each substructure, the first element always denotes the sub-task, which is followed by the arguments and separated by semicolons. Additionally, each element has three components - start token index, end token index, and type of this token span (task/argument type).

Since there is a separate substructure for each sub-task, a particular token span indicating a particular argument can be shared by multiple sub-tasks. For Figure 2, the 'theme' argument is shared by all three sub-tasks, and the 'Source' argument is shared by two sub-tasks. Additionally, this annotation scheme supports a shared argument to be classified differently for different sub-tasks. Similarly, a token or token span can be classified multiple times as different classes, e.g., token 15 is classified as 'Goal' and 'Containing object' is sub-task 'Placing'. The idea behind multiple classifications

is to provide additional information to the task planner.

We have done a separate annotation for argument grounding from an object detector's point of view. If the object classes are changed (in particular the vocabulary set), this annotation has to be changed accordingly. But a separate annotation ensures that the annotation for the task and argument type remains fixed even if the argument grounding annotation is changed. Since an object detector would always recognize an object by the same name, irrespective of it being treated as a different argument type for tasks, one-time prediction is sufficient. Thus, we annotate for argument grounding as a sequence labeling job using the BIO tagging scheme. The BIO tag for the tokens is shown in Figure 2.

|  | Train | Dev | Test |
|---|---|---|---|
| #Instruction | 1,180 | 182 | 580 |
| #single task instruction | 755 | 145 | 417 |
| #multi task instruction | 425 | 37 | 163 |

Table 2: Statistics of the instructions in our dataset.

| Task type | Train | Dev | Test |
|---|---|---|---|
| being_located | 64 | 9 | 16 |
| being_in_category | 49 | 8 | 16 |
| bringing | 160 | 8 | 17 |
| changing_oper._state | 45 | 9 | 12 |
| checking_state | 22 | 9 | 14 |
| cutting | 14 | 7 | 12 |
| following | 52 | 12 | 13 |
| giving | 34 | 8 | 13 |
| inspecting | 10 | 3 | 10 |
| motion | 465 | 42 | 227 |
| opening | 60 | 13 | 17 |
| picking | 259 | 28 | 128 |
| placing | 226 | 28 | 105 |
| pushing | 12 | 7 | 14 |
| rotation | 380 | 34 | 228 |
| searching | 42 | 14 | 24 |
| **#total** | 1,894 | 239 | 866 |

Table 3: Statistics of the different task types in our dataset.

Once prepared, the data was first proportionately divided to fit the training, development, and testing sets with 1,180, 182, and 580 inputs, respectively (See Table 2). We include the statistics about different task types in Table 3. Though the dataset is not balanced in terms of the number of task types, while splitting, we ensured that the distribution of the task types is similar in each of the splits. Also, there is a balance between instructions with single-task and multiple-tasks (Table 2). Additionally, we carefully selected the test set such that there is less

than 60% overall in the input as compared to the train and development set.

## 4.2 Evaluation Metric

We define the performance of our system using a strict F1 score, which is the harmonic mean of the precision and recall of our classifier as the metric definition. To that effect, we consider a missing subtask or attribute label to be negative which means that in the case of no detection or in the case of the wrong classification, the metric takes the prediction as 'wrong'. Conversely, only on a correct label match, it takes the metric as correct. Every hit and every miss is recorded in a confusion matrix where it is populated for all types of tasks and all attributes with 'None' being the record for the misses of a given task or attribute. Thereafter the confusion matrices are used to calculate the precision and the recall for the particular task or attribute classes. Additionally, a combined F1 score is also generated as an overall metric for the baseline comparisons that take into account both the task and attribute combinations together.

## 5 Results

To evaluate our model, we have defined a number of baseline systems. Pramanick et al. (2019a) proposed the first baseline where they have used a CRF-based model for task and argument extraction (Feature CRF in Table 4. Essentially, the model works as a sequence labeling job. Apart from lower accuracy in task and argument prediction, such a model (sequence labeling) cannot – (i) classify the shared task/argument, (ii) reclassify token(s), and (iii) resolve coreference. The next baseline that we used is a pre-trained BERT model and a fully connected layer as a classification head on top. The performance of Feature CRF and BERT baseline is very similar. Then, we combine these two approaches, i.e., instead of using a pre-trained token embedding (like Feature CRF), we used BERT as the encoder layer and the CRF as the classification layer. Though this BERT CRF performs better as compared to the Feature CRF or BERT model, it again emits the behavior of sequence labeling. Also, we have used two sequence-to-sequence models as baseline by finetuning them, namely BART (Lewis et al., 2019) and T5 (Raffel et al., 2020).

As mentioned earlier, tagE follows a generative approach to tackle the limitations of the existing

| Model | Inference time | Without arg grounding | | | With arg grounding | | |
|---|---|---|---|---|---|---|---|
| | | Prec. | Rec. | F1 | Prec. | Rec. | F1 |
| Feature-CRF | 28.9 ms | 0.60 | 0.58 | 0.59 | 0.48 | 0.47 | 0.47 |
| BERT | 91.2 ms | 0.65 | 0.60 | 0.62 | 0.52 | 0.48 | 0.50 |
| BERT-CRF | 102.1 ms | 0.68 | 0.61 | 0.64 | 0.53 | 0.50 | 0.51 |
| BART (with beam width 1) | 341.8 ms | 0.68 | 0.68 | 0.68 | 0.59 | 0.60 | 0.60 |
| BART (with beam width 10) | 754.79 ms | 0.68 | 0.70 | 0.69 | 0.60 | 0.62 | 0.61 |
| T5 (with beam width 1) | 626.72 ms | 0.73 | 0.74 | 0.73 | 0.66 | 0.67 | 0.66 |
| T5 (with beam width 10) | 2029.17 ms | 0.74 | 0.74 | 0.74 | 0.67 | 0.67 | 0.67 |
| tagE | 108.2 ms | 0.85 | 0.80 | **0.82** | 0.72 | 0.67 | **0.69** |

Table 4: Performance of tagE with respect to various baseline methods.

| BERT Encoder | Number of parameters | Training time/epoch | Inference time | F1 without arg grounding | F1 with arg grounding |
|---|---|---|---|---|---|
| mini | 17.5 M | 7.9 s | 77.1 ms | 0.74 | 0.62 |
| small | 44.8 M | 8.5 s | 79.7 ms | 0.76 | 0.64 |
| medium | 57.5 M | 9.6 s | 90.1 ms | 0.78 | 0.66 |
| base | 139.2 M | 13.3 s | 108.2 ms | **0.82** | **0.69** |
| large | 382.4 M | 23.0 s | 135.6 ms | 0.80 | 0.68 |

Table 5: tagE is trained with different-sized BERT encoders, resulting in different parameters and F1 scores.

| Task type | Prec. | Rec. | F1 |
|---|---|---|---|
| being_located | 1.00 | 0.94 | 0.97 |
| being_in_category | 1.00 | 1.00 | 1.00 |
| bringing | 0.94 | 0.67 | 0.78 |
| changing_operational_state | 1.00 | 1.00 | 1.00 |
| checking_state | 1.00 | 0.93 | 0.97 |
| cutting | 0.92 | 1.00 | 0.96 |
| following | 0.85 | 1.00 | 0.92 |
| giving | 1.00 | 1.00 | 1.00 |
| inspecting | 0.20 | 1.00 | 0.33 |
| motion | 0.94 | 0.93 | 0.93 |
| opening | 0.94 | 1.00 | 0.97 |
| picking | 0.92 | 0.96 | 0.94 |
| placing | 0.99 | 0.97 | 0.98 |
| pushing | 0.50 | 1.00 | 0.67 |
| rotation | 0.99 | 0.95 | 0.97 |
| searching | 0.92 | 0.92 | 0.92 |

Table 6: Task-wise performance of tagE.

| Argument type | Prec. | Rec. | F1 |
|---|---|---|---|
| agent | 0.85 | 0.80 | 0.82 |
| area | 0.54 | 0.34 | 0.42 |
| category | 0.17 | 0.20 | 0.18 |
| container_portal | 0.76 | 0.81 | 0.79 |
| containing_object | 0.70 | 0.76 | 0.73 |
| cotheme | 0.77 | 0.91 | 0.83 |
| degree | 0.52 | 0.89 | 0.65 |
| desired_state | 0.43 | 0.46 | 0.44 |
| device | 0.77 | 0.83 | 0.80 |
| goal | 0.85 | 0.85 | 0.85 |
| cogoal | 0.74 | 0.81 | 0.78 |
| manner | 0.89 | 0.89 | 0.89 |
| operational_state | 0.67 | 0.67 | 0.67 |
| recipient | 1.00 | 1.00 | 1.00 |
| source | 0.71 | 0.74 | 0.72 |
| cosource | 0.76 | 0.81 | 0.79 |
| theme | 0.88 | 0.87 | 0.87 |

Table 7: Argument-wise performance of tagE.

methods. The performance gain is evident from Table 4. We have calculated a combined F1 score considering all the sub-tasks and the corresponding arguments in an input. As the argument grounding is separately annotated, this training can be done separately. We have done two sets of training – one with the argument grounding training as a separate job, and another with a joint job of argument grounding and task & argument type prediction. The experiment shows that joint training performs

much better than separate training.

As ablation, we have experimented with different types of BERT encoders. Though all types of BERT models use a number of stacked transformer layers to get a vector representation of natural language input, the variation stems from the number of transformer layers and dimension of the output vectors. We have experimented with 5 different variations – (i) mini with 4 layers and

256-dimensional vector, (ii) small with 4 layers and 512-dimensional vector, (iii) medium with 8 layers and 512-dimensional vector, (iv) base uncased with 12 layers and 786-dimensional vector, and (v) large uncased with 24 layers and 1024-dimensional vector. The pre-trained models for the variations of BERT is provided by Turc et al. (2019). A small number of layers and vector dimensions leads to less number of parameters, a smaller model size, and a smaller training & inference time. However, this impacts the accuracy of the system as evident in Table 5. With a larger encoder network, the performance of tagE keeps increasing with some saturation point. The large BERT model, even though it has a much larger network, is unable to outperform the BERT base model. Thus, we fixated on tagE with the BERT base model as the encoder. We provide further accuracy statistics of each tasks and arguments for the test split of the dataset for tagE. The details are summarized in Table 6 and 7.

## 6 Conclusions

Instructing a robot in natural language certainly improves the usability and acceptability of the robot. However, understanding the instructions given in unstructured text can be challenging for a robot. A robot needs to perform multiple jobs to completely understand a natural language instruction such as extracting the (sub)tasks, extracting their associated arguments, and grounding of the arguments. Previously, researchers have tried to solve these jobs independently, thus misses the interaction among them in the given instruction. To overcomes this limitation, we propose a neural network model tagE, which can solve these three tasks together in an end-to-end fashion. We also annotate a suitable dataset for experiments and our proposed approach outperforms some strong baselines on this dataset. In future, we would integrate a planner module that would generate an executable plan from the output of tagE.

## 7 Limitations

In our experiments, we observe that while the model is mostly accurate on the test dataset, it fails on certain long natural language inputs. In particular, the sub-tasks and their arguments are not often not predicted for very long input with more than 5 sub-tasks. However, such a long input with as many sub-tasks is not very common in typical human-robot interactive scenarios. Also,

tagE is trained to extract the sub-tasks in the same sequence as they appear in the instructions. In certain situations, the robot may need to perform the sub-tasks in a different order. For example, in the instruction - "bring a can of beer, if you can find a chilled one", even though the Bringing sub-task appears earlier than the Check-state sub-task, the robot has to perform them in the reverse order. However, these problems are typically solved by task planning.

## 8 Ethics Statements

There is no ethical concerns about this work.

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

# 9 Appendix

We include the example of different task types, argument types, and instructions in Table 8. Each

| Task type | Argument types | Example instruction |
|---|---|---|
| 1. being_located (is) | source (table) | the cup is on the table |
| 2. being_in_category (is) | theme (living room), category (with green curtains) | this is a living room with green curtains |
| 3. bringing (bring) | recipient (me), theme (cup), source(table) | bring me a cup from the table |
| 4. changing_operational_state (turn) | operational_state (on), device (television) | turn on the television |
| 5. checking_state (check) | theme (stereo), desired_state (on) | please check if the stereo is on |
| 6. cutting (cut) | theme (apple), source (dining table) | cut the apple on the dining table |
| 7. following (follow) | cotheme (person), goal (kitchen) | follow the person to the kitchen |
| 8. giving (pass) | theme (plate), agent (robot), recipient (me) | robot can you pass me a plate |
| 9. inspecting (look) | manner (down), source (floor) | look down on the floor |
| 10. motion (go) | goal (window) | go near the window |
| 11. opening (open) | container_portal (cabinet) | open the cabinet |
| 12. picking (take) | theme (bottle), source (bedside table) | take the bottle from the bedside table |
| 13. placing (put) | theme (bottle), goal (trash) | put the bottle on the trash |
| 14. pushing (push) | theme (box), agent (you), source (table) | can you push the box on the table |
| 15. rotation (turn) | manner (your left), agent (robot) | robot turn to your left |
| 16. searching (find) | recipient (me), theme (red shirt) | find me the red shirt |

Table 8: Task and argument types in our dataset with example instruction – the first two tasks are not real tasks. The phrase that represents the task and argument in each instruction are shown inside the bracket.

task and argument corresponds to one/multiple words in the instruction. The corresponding words for the task and arguments are also marked (within the parenthesis).