# OpenReview forum: "tagE: Enabling an Embodied Agent to Understand Human Instructions"
_EMNLP/2023/Conference — EMNLP 2023 Findings_

### Official Review · Reviewer_MBSB · 2023-07-31

**Soundness:** 4

**Excitement:**

3: Ambivalent: It has merits (e.g., it reports state-of-the-art results, the idea is nice), but there are key weaknesses (e.g., it describes incremental work), and it can significantly benefit from another round of revision. However, I won't object to accepting it if my co-reviewers champion it.

**Paper Topic And Main Contributions:**

The paper proposes a model to improve the alignment between the NLU component and the physical environment of a robot. This is realised as grounding tasks (intents) and arguments (objects) extracted from natural language instructions to the representations of the robot's environment.

The main contributions are a neural network-based architecture called GATE to extract intents and arguments from instructions and the adaptation of existing datasets to instantiate the proposed framing for the task.

**Questions For The Authors:**

- What is K in your dataset? Why is K = 2*(#objects) +2, instead of +1 (the O label)?
- Were the two robotic commands datasets combined? How are they related?
- If the paper is accepted, will the code and data be available?

**Reasons To Accept:**

- The paper proposes an end-to-end model that integrates multiple subtasks that ground NLU to the robot's environment representation (sequence labelling for objects, task type and span prediction, argument type and span prediction), with a detailed description of the implemented architecture.
- Two existing datasets are adapted into a new format that allows joint training of these tasks.
- Five baseline models are used for comparison of the performance of the new model.

**Reasons To Reject:**

- Not detailed information on the annotation process in included (was it done programatically? if yes, how? if not, who did it? how many annotators?). Some of the basic statistics that would be useful to understand the data are not reported (number of objects, number of instances per task type, whether there is overlap between the two datasets). The text explaining how the evaluation metric is computed is not very clear.
- The results+discussion section is too short and compressed. Error analysis is minimal, and the ablation only consists in varying the BERT encoder architecture. It would be useful to separately report the performance of each component (the object labelling, the task classification, the task span, the argument classification, the argument span), on each subtask, on each dataset.
- The paper needs proofreading (see some examples of typos, unclear passages and incomplete references below). The limitation section text is duplicated.

**Reproducibility:**

3: Could reproduce the results with some difficulty. The settings of parameters are underspecified or subjectively determined; the training/evaluation data are not widely available.

**Reviewer Confidence:**

3: Pretty sure, but there's a chance I missed something. Although I have a good feel for this area in general, I did not carefully check the paper's details, e.g., the math, experimental design, or novelty.

**Typos Grammar Style And Presentation Improvements:**

Clarifications and unclear or imprecise passages
- [046] I don't think that 'simplified' the tasks is a suitable verb here
- The first two subsections in the related work section are not very informative, as they contain lists of citations without further discussion or how exactly this paper fits in the current context.
- [l451] Which vertical lines?
- [l496] Overall what?
- [l525] Job does not sound as a proper term in this section.
- [538] What does it mean to emit the behaviour of sequence labelling?
- [l606] What does it mean 'not often not predicted'?


Typos and Grammar:
- [l032] are often engage?
- [l061] actions
- [l102] extract
- [l201, l206]
- [l276] the inner
- [l277] responsible for generating
- [l288] refer
- [l448] located
- [l523] closing parenthesis missing
- [l553] & not appropriate here
- [l592] who misses?
- [l599] In the future

Style

- For the paragraph titles in Section 2, I suggest using the command \paragraph{}.
- When the citation is a part of the text, use \citet instead of \citep (for example, lines 144, 147, 150).
- Do not type the citation right before the citation itself (e.g. in l209, l216, l222, l224).
- Suggestion: use {} for the set notation.
- Algorithm 1 has no reference in the text.
- Please check the APA guideliens for table formatting. In particular, horizontal and vertical lines should be avoided. For instance, horizontal lines are only needed at the top and bottom and after the header.
- Some references are incomplete (e.g. Bahdanau 2015, Devlin 2019, Kundu 2018). Besides, as per ACL guideliens, arXiv versions should not be cited when the paper has been published in a venue (e.g. Blukis 2019, Huang 2022)

---

> ### Author Rebuttal · Authors · 2023-08-29
>
> We are grateful for the reviewer’s constructive comments and suggestions. We have addressed them in the following.
>
> **Q1**: Not detailed information on the annotation process in included (was it done programmatically? if yes, how? if not, who did it? how many annotators?).
>
> **Response**: The annotation is done in two stages. First, a group of junior annotators created the dataset including the instruction and annotation. A common schema is defined for the annotation. In the next stage, a senior annotator checked all the annotations carefully. The instructions were carefully chosen so that there is sufficient natural variation and does not does vary in one/two tokens.
>
> **Q2**: Some of the basic statistics that would be useful to understand the data are not reported (number of objects, number of instances per task type, whether there is overlap between the two datasets).
>
> **Response**: We will include such statistics about our dataset in the final version and will also release it for future research.
>
> **Q3**: The text explaining how the evaluation metric is computed is not very clear. The results+discussion section is too short and compressed. Error analysis is minimal, and the ablation only consists in varying the BERT encoder architecture. It would be useful to separately report the performance of each component (the object labeling, the task classification, the task span, the argument classification, the argument span), on each subtask, on each dataset.
>
> **Response**: Yes, we should have done better space management to include all these  results. For example,
> Task-wise accuracy looks like this –
>
> [Being_located - 0.968,  Being_in_category - 0.769, Bringing - 0.97, Changing_operational_state - 1.0, Checking_state - 1.0,  Closure - 0.903, Cutting - 1.0, Following - 1.0, Giving - 1.0, Inspecting - 0.182, Motion - 0.964, Placing - 1.0, Picking - 0.972, Pushing - 0.981, Rotation - 0.959, Searching 0.737]
>
> In the final version, there is a provision of one extra page, which we shall utilize to describe the dataset and detailed results.
>
> **Q4**: The paper needs proofreading (see some examples of typos, unclear passages and incomplete references below). The limitation section text is duplicated.
>
> **Response**: We thank the reviewer for pointing these out. We will address all the typos, grammar, clarity and citation issues in the final version of the paper. We have fixed the text duplication in the limitation section.
>
>
> **Q5**: What is K in your dataset? Why is K = 2*(#objects) +2, instead of +1 (the O label)?
>
> **Response**: K is the total number of labels for BIO tagging. Hence, K should be 2*(#objects) + 1.
>
> **Q6**: Were the two robotic commands datasets combined? How are they related?
>
> **Response**: Yes, we have combined the two datasets. However, we have selected a subset of the text instructions from both the dataset setting the constraint that each instruction should contain task types from the task-set mentioned in the paper.
> Both the datasets are for evaluation of a robotic agent to perform tasks given in natural language instructions.
>
> Please note, we have added around 40% instructions on top of these combined dataset and annotated the entire set.
>
> **Q7**: If the paper is accepted, will the code and data be available?
>
> **Response**: We shall definitely release the dataset along with all the necessary details. We shall also add a few more details of the dataset, e.g., task distribution. We may not be able to release the full code due to our organization’s policy. However, we shall help any researchers in reproducing the code and result. We shall add a line in the article regarding this.
>
> **Q8**: Typos Grammar Style And Presentation Improvements: Clarifications and unclear or imprecise passages [046] I don't think that 'simplified' the tasks is a suitable verb here The first two subsections in the related work section are not very informative, as they contain lists of citations without further discussion or how exactly this paper fits in the current context. [l451] Which vertical lines? [l496] Overall what? [l525] Job does not sound as a proper term in this section. [538] What does it mean to emit the behaviour of sequence labelling? [l606] What does it mean 'not often not predicted'? Typos and Grammar: [l032] are often engage? [l061] actions [l102] extract [l201, l206] [l276] the inner [l277] responsible for generating [l288] refer [l448] located [l523] closing parenthesis missing [l553] & not appropriate here [l592] who misses? [l599]
>
> **Response**:
>
> [046] Agreed. We have changed the text segment to “...large scale language models have shown significant improvements in many NLP tasks (Devlin et al., 2019; Brown et al., 2020).”
>
> We will make the first two subsections of related work more clear by discussing the relevance of the cited papers with our work.
>
> [l451] We overlooked this mistake in description. To clarify, we earlier visualized the different tasks in the example in Figure 2 with vertical lines, which we later removed but did not correct the text accordingly.
>
> [l496] This was a typo. We meant to say overlap instead of overall. We calculate the overlap in terms of word-level lexical similarity.
>
> [l525] Agreed, we used the term to avoid confusion with the term ‘task’. We have changed the text segment to “.. the model performs sequence labeling”
>
> [538] We meant to say that this models also has the same limitations of a sequence labeling model. We’ll rephrase this text in the final version.
>
> [l606] We mean to say that the decoder sometimes predicts lesser number of tasks and the corresponding arguments compared to the reference annotation, for very long inputs.
>
> [l032] - [l599] We shall fix the typos and grammar mistakes.
>
>
> **Q9**: In the future Style For the paragraph titles in Section 2, I suggest using the command \paragraph{}.
>
> **Response**: We’ll fix this in the final version of the paper.
>
> **Q10**: When the citation is a part of the text, use \citet instead of \citep (for example, lines 144, 147, 150). Do not type the citation right before the citation itself (e.g. in l209, l216, l222, l224). Suggestion: use {} for the set notation.
>
> **Response**: We will do the needful.
>
> **Q11**: Algorithm 1 has no reference in the text. Please check the APA guideliens for table formatting. In particular, horizontal and vertical lines should be avoided. For instance, horizontal lines are only needed at the top and bottom and after the header.
>
> **Response**: We will take care of it.
>
> **Q12**: Some references are incomplete (e.g. Bahdanau 2015, Devlin 2019, Kundu 2018). Besides, as per ACL guideliens, arXiv versions should not be cited when the paper has been published in a venue (e.g. Blukis 2019, Huang 2022)
>
> **Response**: Noted; we shall add proper citation.

---

### Official Review · Reviewer_Tmep · 2023-08-06

**Typos Grammar Style And Presentation Improvements:** NA
**Soundness:** 3

**Excitement:**

3: Ambivalent: It has merits (e.g., it reports state-of-the-art results, the idea is nice), but there are key weaknesses (e.g., it describes incremental work), and it can significantly benefit from another round of revision. However, I won't object to accepting it if my co-reviewers champion it.

**Missing References:**

NA

**Paper Topic And Main Contributions:**

This paper curates a new dataset that enables grounded task-argument extraction, and proposes an encoder-decoder based NLU for Embodied Agents.

**Questions For The Authors:**

Same as Reasons To Reject

**Reasons To Accept:**

This paper curates a new dataset that enables grounded task-argument extraction.

In this scenario of Embodied Agents, this paper proposes to ground on the argument and the task simultaneously, and designs an encoder-decoded architecture that employs layered decoding.

**Reasons To Reject:**

The dataset contains less than 2k examples, which may be too small.

In the experiment, the baseline models only contains CRF, BERT and T5. However, in order to demonstrate the effectiveness of the proposed model, it is beneficial to request results from ChatGPT, by constructing appropriate prompts.

**Reproducibility:**

3: Could reproduce the results with some difficulty. The settings of parameters are underspecified or subjectively determined; the training/evaluation data are not widely available.

**Reviewer Confidence:**

4: Quite sure. I tried to check the important points carefully. It's unlikely, though conceivable, that I missed something that should affect my ratings.

---

> ### Author Rebuttal · Authors · 2023-08-29
>
> We appreciate the reviewer’s insightful feedback and recommendations. Our replies are given below.
>
> **Q1**: The dataset contains less than 2k examples, which may be too small.
>
> **Response**: Even though we have a relatively smaller dataset, it contains quite a lot of natural variation and combinations of multiple tasks. We have noticed that now we can add more instructions that would increase the absolute count, but not the variety in the instruction.
>
> **Q2**: In the experiment, the baseline models only contains CRF, BERT and T5. However, in order to demonstrate the effectiveness of the proposed model, it is beneficial to request results from ChatGPT, by constructing appropriate prompts.
>
> **Response**: We believe that BERT-CRF, BART, and T5 are strong baselines considering the application premise. We have to run this module along with a few more neural network models, like ASR, Vision, TTS for a robotics application, where network connectivity is not always available. These models have to run on edge device (or on-board). Thus, memory and processing power is a concern. Also, the overall system output has a time budget (real-timeness). Therefore, utilizing chatGPT is not feasible for our application.
>
> **Q3**: Could reproduce the results with some difficulty. The settings of parameters are underspecified or subjectively determined; the training/evaluation data are not widely available.
>
> **Response**: We shall definitely release the dataset along with all the necessary details. We shall also add a few more details of the dataset, e.g., task distribution. We may not be able to release the full code due to our organization’s policy. However, we shall help any researchers in reproducing the code and result. We shall add a line in the article regarding this.

---

### Official Review · Reviewer_vT9w · 2023-08-06

**Soundness:** 2

**Excitement:**

3: Ambivalent: It has merits (e.g., it reports state-of-the-art results, the idea is nice), but there are key weaknesses (e.g., it describes incremental work), and it can significantly benefit from another round of revision. However, I won't object to accepting it if my co-reviewers champion it.

**Missing References:**

Wu, Zhenyu, et al. "Embodied Task Planning with Large Language Models." arXiv preprint arXiv:2307.01848 (2023).
https://arxiv.org/pdf/2307.01848
might be worth taking into account in "Related Work".

**Paper Topic And Main Contributions:**

An encoder-decoder architecture is presented that learns to extract subtasks and related arguments from compound natural language task instructions. The extracted tasks are mapped to the known skillset of a robot, the arguments to objects in the environment. Apart form the proposed neural model, a dataset for task-argument extraction is presented. However, it is unclear whether the authors intend to publish the dataset.

**Reasons To Accept:**

The paper presents a neural approach and respective dataset to extracting subtasks and their arguments from complex NL instructions for robots. This is quite helpful for natural human-robot interaction.
The dataset should be published.

**Reasons To Reject:**

The presentation of the evaluation results lacks some detail. See “Presentation Improvements”.

**Reproducibility:**

3: Could reproduce the results with some difficulty. The settings of parameters are underspecified or subjectively determined; the training/evaluation data are not widely available.

**Reviewer Confidence:**

4: Quite sure. I tried to check the important points carefully. It's unlikely, though conceivable, that I missed something that should affect my ratings.

**Typos Grammar Style And Presentation Improvements:**

Line 140: briefly explain what B, I and O stands for
Line 448: typo “locatede”
Line 449-452: “Each subtask … example.” Unclear what it refers to. Seems to be a remnant from a previous version of the paper.
Line 610-613: “In our … inputs.” is a duplication of line 603-605. Delete.
Table 3: does “score without arg grounding” mean that task- and argument training is done separately? If yes, there is an inconsistency with what is said in line 554-555. Please, clarify.
Section 4.1: When describing the dataset, add information on the distribution of single task and multi task instances relative to task types. Are the instances available per type approximately balanced?
As regards the evaluation, it is unclear whether the F1 scores relate to single train-test runs or to n-folds. In any case, report results from n-folds. Also do an error analysis for the GATE model separating results for single and multiple tasks and task types.
Note: There are a few singular/plural and subject-verb incongruencies throughout the text.
Note that GATE in NLP is a well known acronym, see https://gate.ac.uk.
Reconsider your ethics statement: just to give an example, when the model is used with real robots in real-world scenarios, performance quality matters. Therefore, users should be well aware of system limitations.

---

> ### Author Rebuttal · Authors · 2023-08-29
>
> First of all, we thank the reviewer for the invaluable comments and suggestions. Please find our responses in the following.
>
> **Q1**: Missing References: Wu, Zhenyu, et al. "Embodied Task Planning with Large Language Models." arXiv preprint arXiv:2307.01848 (2023). https://arxiv.org/pdf/2307.01848 might be worth taking into account in "Related Work".
>
> **Response**: Thank you for pointing to this related work. As this was published in Arxiv after the submission deadline, we were not aware of this work. We shall include this as part of the related work.
>
> **Q2**: Typos Grammar Style And Presentation Improvements
>
> **Response**: We are sorry for not paying greater attention to the proof reading and thankful to the reviewer for pointing these. We will address all the typos, grammar styles, presentation limitation, etc. in the final version.
>
> **Q3**: briefly explain what B, I and O stands for
>
> **Response**: In the BIO tagging scheme, B, I, O stands for beginning, inside, and outside respectively. We shall include this in the final version of the article with a pointer to the example given in Figure 2.
>
> **Q4**: “Each subtask … example.” Unclear what it refers to.
>
> **Response**: We have incorrectly used the term subtask. It is only tasks that we are referring to. There can be complex and compound instruction, and the instruction can have multiple tasks as listed Figure 4. We shall replace all the mention of subtask with task.
>
> **Q5**: “In our … inputs.” is a duplication of line 603-605. Delete
>
> **Response**: Thank you for pointing it out. We shall do it.
>
> **Q6**: Table 3: does “score without arg grounding” mean that task- and argument training is done separately? If yes, there is an inconsistency with what is said in line 554-555. Please, clarify.
>
> **Response**: There are three parts of the network – argument (object) grounding, task prediction, argument prediction. Training for task and argument prediction are always done together. If you look at Figure 2, the “annotation” row includes task and argument prediction along with their span is always predicted together. Now, for the “Placing” task, there is “Goal” argument. The token “fridge” is predicted as the “goal”. “Score without arg grounding” includes accuracy for the task, corresponding args and the span is matching with the ground truth. This does not include the accuracy of predicting (grounding) this “fridge” as “Refrigerator”. This argument grounding is necessary as we may refer the objects in different way. But, the object detector that the robot uses should be able to map the objects to the vision algorithm’s vocabulary. This is done with BIO tagging and later at post-processing matching the span.
>
> We shall clarify this in the final version of the article.
>
> **Q7**: Section 4.1: When describing the dataset, add information on the distribution of single task and multi task instances relative to task types. Are the instances available per type approximately balanced?
>
> **Response**: We have provided an overall details in Table 2. We will include further break-ups of the dataset in the final version.
>
> **Q8**: As regards the evaluation, it is unclear whether the F1 scores relate to single train-test runs or to n-folds. In any case, report results from n-folds.
>
> **Response**: The current results depicts the F1 score of a single test score run as depicted in table 3 & 4. We have done train-test with 5 different seeds. The resulting F1 scores are – “without arg grounding”, it is 0.78 with 0.03 standard deviation (it was reported 0.79 in the paper);  “with arg grounding”, it is  0.66 with standard deviation 0.028 (it was reported 0.67 in the paper).
> We shall replace all the results with average on multiple train-test runs.
>
> **Q9**: Also do an error analysis for the GATE model separating results for single and multiple tasks and task types.
>
> **Response**: We will get one extra page in the final version and use it for such analysis.
>
> **Q10**: Note: There are a few singular/plural and subject-verb incongruencies throughout the text.
>
> **Response**: As mentioned earlier, we shall do a thorough proof reading.
>
> **Q11**: Reconsider your ethics statement: just to give an example, when the model is used with real robots in real-world scenarios, performance quality matters. Therefore, users should be well aware of system limitations.
>
> **Response**: We agree with this. We shall add an ethics statement after deliberation with our ethical committee.
>
> **Q12**: The contribution depends on data that are simply not available outside the author's institution or consortium; not enough details are provided.
>
> **Response**: We shall definitely release the dataset along with all the necessary details. We shall also add a few more details of the dataset, e.g., task distribution. We may not be able to release the full code due to our organization’s policy. However, we shall help any researchers in reproducing the code and result. We shall add a line in the article regarding this.

---

### Meta-Review · Area_Chair_bKx8 · 2023-09-19

**Recommendation:** 3

**Metareview:**

This paper proposes a system for resolving and following compound (multi-task) instructions in a robot instruction following task. The approach also maps the extracted tasks and arguments to known robot skills and the surrounding environment. The paper proposes a formal annotation scheme to represent subtasks and referenced objects. Performance is quantified using label prediction accuracy (rather than downstream instruction following performance), as well as inference time as the end goal is to run inference on-device. Details on the annotation process, dataset statistics, and more analysis of model errors should be added to the next revision of the paper.

---

### Decision · Program_Chairs · 2023-10-07

**Decision:**

Accept-Findings

**Comment:**

This paper proposes a system for resolving and following compound (multi-task) instructions in a robot instruction following task. The approach also maps the extracted tasks and arguments to known robot skills and the surrounding environment. The paper proposes a formal annotation scheme to represent subtasks and referenced objects. Performance is quantified using label prediction accuracy (rather than downstream instruction following performance), as well as inference time as the end goal is to run inference on-device. Details on the annotation process, dataset statistics, and more analysis of model errors should be added to the next revision of the paper.